# Identification of a Novel Frameshift Variant in *MYF5* Leading to External Ophthalmoplegia with Rib and Vertebral Anomalies

**DOI:** 10.3390/genes15060699

**Published:** 2024-05-27

**Authors:** Paulina Ocieczek, Ngozi Oluonye, Cécile Méjécase, Elena Schiff, Vijay Tailor, Mariya Moosajee

**Affiliations:** 1Moorfields Eye Hospital NHS Foundation Trust, London EC1V 9EL, UK; p.ocieczek@nhs.net (P.O.);; 2UCL Institute of Ophthalmology, London EC1V 9EL, UK; 3Great Ormond Street Hospital for Children NHS Foundation Trust, London WC1N 9JH, UK; 4Francis Crick Institute, London NW1 1AT, UK

**Keywords:** MYF5, myogenic factor 5, external ophthalmoplegia with vertebral and rib anomalies (EORVA), external ophthalmoplegia, scoliosis, rib anomalies, myogenic transcription factors

## Abstract

Myogenic transcription factors with a basic helix–loop–helix (bHLH) such as MYOD, myogenin, MRF4, and MYF5 contribute to muscle differentiation and regulation. The *MYF5* gene located on chromosome 12 encodes for myogenic factor 5 (MYF5), which has a role in skeletal and extraocular muscle development and rib formation. Variants in *MYF5* were found to cause external ophthalmoplegia with rib and vertebral anomalies (EORVA), a rare recessive condition. To date, three homozygous variants in *MYF5* have been reported to cause EORVA in six members of four unrelated families. Here, we present a novel homozygous *MYF5* frameshift variant, c.596dupA p. (Asn199Lysfs*49), causing premature protein termination and presenting with external ophthalmoplegia, ptosis, and scoliosis in three siblings from a consanguineous family of Pakistani origin. With four *MYF5* variants now discovered, genetic testing and paediatric assessment for extra-ocular features should be considered in all cases of congenital ophthalmoplegia.

## 1. Introduction

During mammalian development, skeletal muscle differentiation is regionalised with skeletal muscles of the trunk, limbs, diaphragm, and tongue derived from somites, while the craniofacial muscles, including extraocular muscles (EOM), originate from prechordal and paraxial mesoderm [1]. In the case of disease or injury, skeletal muscles can be regenerated by myogenic precursor cells called satellite cells [2,3]. EOM are more complex than limb skeletal muscles with smaller motor units, higher mitochondrial content, increased blood flow from a dense vascular bed, and co-expression of several myosin isoforms. They are the fastest contracting muscles in the human body and are highly resistant to fatigue [4,5].

Embryonic and post-natal myogenesis of skeletal muscle and EOM is regulated by four basic helix–loop–helix (bHLH) transcription factors: MYF5, MYOD, MYOG (also called myogenin), and MRF4 (also called MYF6 or herculin) [6,7,8,9,10]. MYF5 is the first myogenic factor to be expressed; in mice, *Myf5* expression occurs around embryonic day 8 (E8), preceding somite differentiation into the dermis, axial muscles, vertebrae, and ribs [11]. *Myf5* or *Mrf4* activates myogenesis via *MyoD* expression, which initiates *MyoG* expression [12,13,14,15,16] (Figure 1).

Disruption of the *Myf5* gene in mice causes abnormal development of the distal parts of the ribs and postnatal death due to respiratory distress [15]. Studies on mice carrying heterozygous variants in *Myf5* in *trans* with a second heterozygous variant in *Mrf4* (*Myf5^+/m1^ Mrf4^+/bh1^*) showed severe rib anomalies and undetectable myotome formation, similar to *Myf5* knockout models [17]. Homozygous *Mrf4* mutants also exhibit rib defects, but the most severe rib anomalies are present in *Myf5* null models [17,18]. *Myf5* mutant mice with normal expression of *Myod1*, *MyoG,* and *Mrf4* develop normal skeletal muscles, but are lethal due to an inability to breathe [15]. Inserting *MyoG* cDNA into the *Myf5* locus via homologous recombination leads to partial phenotypic rescue, with development of a normal rib cage in *MyoG* knock-in mice [19]. In mice lacking both MYF5 and MYOD transcription factors, no skeletal muscles were formed, and mice died postnatally within minutes from birth. The presence of a healthy copy of either *MyoD* or *Myf5* in mutant mice led to partial or full skeletal muscle development [7].

Congenital fibrosis of extraocular muscles (CFEOM) characterises a non-progressive congenital ophthalmoplegia with or without ptosis. It is a development disorder primarily affecting cranial nerves (a cranial innervation disorder) and results in fibrosis and hypoplasia of innervated EOMs. It can be inherited in an autosomal dominant or autosomal recessive manner and there are five types; types 1 and 3 are autosomal dominant, whilst types 2, 4, and 5 are inherited recessively. Six genes are known to be associated with this condition: *COL25A1*, *KIF21A2*, *PHOX2A*, *TUBA1A*, *TUBB2B,* and *TUBB3*. Systemic features can present as neurodevelopmental, brain, or limb anomalies, including reduced numbers of fingers or toes (oligodactyly).

External ophthalmoplegia with rib and vertebral abnormalities (EORVA) is a rare autosomal recessive disease associated with *MYF5* variants and is characterised by congenital, non-progressive ophthalmoplegia and ptosis, with vertebral and rib anomalies, scoliosis, and torticollis. First described in 2018, Di Goia et al. reported three families presenting with this condition caused by biallelic mutations in the *MYF5* gene (OMIM # 159990) located on 12q21.31 [20]. Unlike CFEOM, the primary pathology is not neurological but originates in the muscle. The characteristic ocular features are similar in both EORVA and CFEOM. Overlooking mild systemic features that differ in both these conditions can lead to misdiagnosis and suboptimal management (Table 1). Ocular management in EORVA and CFEOM include monitoring and correction of refractive error and amblyopia; in some cases, surgery to correct ptosis or extraocular muscles alignment should be considered. Management of systemic complications will vary depending on presenting features and include multidisciplinary management and the involvement of a paediatrician.

Herein, we describe a novel homozygous c.596dupA variant in the *MYF5* gene associated with EORVA in three siblings from a consanguineous family of Pakistani origin.

## 2. Case Description

Two brothers aged 8 (IV-1) and 7 years (IV-2), and their younger sister aged 4 years (IV-3) from a consanguineous family of Pakistani origin were referred to the genetic eye disease clinic at Moorfields Eye Hospital (MEH) with a diagnosis of CFEOM. Their parents were unaffected (Figure 2).

Patient IV-1 first presented to a general paediatric ophthalmology service aged six months with left exotropia, left hypertropia, and ophthalmoplegia. Magnetic resonance imaging (MRI) of the head and orbits showed smaller left medial rectus, superior rectus and superior oblique muscles compared to the contralateral side, leading to a presumed diagnosis of CFEOM (Figure 3J). At most recent presentation, age 8 years, his ophthalmic examination showed a best corrected visual acuity (BCVA) of 0.00 LogMAR in the right and 0.30 LogMAR in the left eye (Table 2). Orthoptic assessment showed constant left exotropia, bilateral ophthalmoplegia, chin elevation, and mild bilateral ptosis. Anterior and posterior segment examination showed no abnormalities. Optos imaging and optical coherence tomography (OCT) of the macula showed no abnormalities (Figure 3M).

Patient IV-2 presented to a general paediatric ophthalmology clinic at 23 months with ophthalmoplegia. There were no notable concerns regarding intellect or development. At age 7 years, his ocular examination showed BCVA of 0.00 LogMAR in the right and 0.08 LogMAR in the left eye, a constant right alternating exotropia, bilateral ophthalmoplegia, and mild bilateral ptosis (Table 2). Anterior and posterior segment examination was within normal limits.

Patient IV-3 was originally found to have restricted eye movements aged four months. She had normal intellect but there were some minor concerns about clumsiness; her neurological examination was unremarkable but mild in-toeing was noted. At age 4 years BCVA was 0.24 LogMAR in each eye. An intermittent distance left exotropia was detected with bilateral ophthalmoplegia and mild right-sided ptosis (Table 2). Anterior and posterior segment examination was normal, Optos imaging and OCT scans were within normal limits.

Previously reported variants in *MYF5* were known to cause external ophthalmoplegia, with vertebral and rib anomalies [18]. Therefore, following the genetic results, patients IV-1, IV-2, and IV-3 were assessed by a developmental paediatrician to look for syndromic features, in particular torticollis, scoliosis, spinal, and rib cage or chest abnormalities, none of which were found on physical examination. Neurological examination and growth indices were within normal limits. Patients were referred for spinal X-rays to investigate for radiological evidence of ribcage and spinal abnormalities. Varying degrees of thoracic, thoracolumbar, and lumbar scoliosis were reported in all three patients (Figure 3C,D).

## 3. Genetic Testing

Informed consent was obtained from all subjects involved in the study through the Genetic Study of Inherited Eye Disease (REC reference 12/LO/0141). A clinical exome (Agilent SureSelect Focused Exome +1 capture) for Patient IV-1 was performed on the Illumina NextSeq 500 platform, with sequence data generated across the full capture region of greater than 5000 genes. Next-generation sequencing analysis was then performed for a virtual panel of coding exons (+/− 20 bp) of 14 genes associated with eye movement disorders (EMD_v2 panel, North East Thames Regional Genetics Laboratory: *CHN1, COL25A1, DCC, FRMD7, HOXA1, HOXB1, KIF21A, MAFB, PHOX2A, ROBO3, SALL1, SALL4, TUBB2B, TUBB3*). Larger insertion/deletion mutations and copy variants were analysed using ExomeDepth. Variants were filtered according to minor allele frequency (>2%) from 1000G, ExAC or EVS databases.). No pathogenic or likely pathogenic variants were identified. Variants in non-coding regions that could affect gene expression could not be excluded.

Both brothers IV-1 and IV-2 were subsequently recruited to the Genomics England 100,000 Genomes Project together with their unaffected parents for whole genome sequencing as previously described [21,22]. Both brothers were found to be homozygous, and the unaffected parents heterozygous carriers for a novel duplication c.596dupA in the *MYF5* gene resulting in a frameshift variant p.(Asn199Lysfs*49). Targeted sequence analysis of the *MYF5* variant confirmed the 100,000 Genome Project findings. The third affected sibling, IV-3, underwent familial *MYF5* testing and was found to be homozygous for the same variant. In the gnomAD population database, this variant was found at heterozygous state in two individuals (f = 0.00000124) and has not been described before at homozygous state.

According to ACMG variant classification guidelines, *MYF5* c.596dupA is likely pathogenic. It is a null variant in a gene where loss of function is the presumed mechanism of disease and the variant results in a reduction of more than 10% of the protein (PVS1 strong). The variant is present in only two alleles (no homozygotes) in the gnomAD v4.0 database (PM2 moderate). In vitro studies would help confirm if any residual function of the truncated protein is present and would help to upgrade the classification.

## 4. Discussion

Here we present the three siblings with EORVA (non-progressive ophthalmoplegia, ptosis, and scoliosis), without vertebrae anomalies and torticollis, caused by a novel homozygous *MYF5* frameshift variant, c.596dupA, p.(Asn199Lysfs*49), in exon 3. Three *MYF5* variants have been previously reported to be associated with EORVA in six members of four unrelated families: (i) deletion c.23_32del p.(Gln8Leufs*86); (ii) deletion c.191del p.(Ala64Valfs*33); and (iii) missense variant c.283C>T p.(Arg95Cys), all located in exon 1 (Figure 4, Table 3) [18,21].

The homozygous 10bp deletion, c.23_32del, in exon 1 was reported in a brother, age 9 years, and his sister, age 19 years, of Turkish descent from unaffected consanguineous first cousins. Both exhibited external ophthalmoplegia, ptosis, squint, scoliosis, torticollis, and dysmorphic hypoplastic ribs. The same variant c.23_32del was discovered in another 16-year-old male from the same village, with similar ocular and vertebral features, and pectus carinatum [20]. The c.23_32del variant appears to exert a more severe extra-ocular phenotype compared to our reported c.596dup variant, where scoliosis was the only skeletal feature detected on X-ray imaging. The c.191del variant in exon 1 was found in an 8-year-old Chinese boy with paternal uniparental disomy. Extra-ocular features included scoliosis and hypoplastic ribs. This patient had a milder ocular phenotype with only ptosis reported: however, the ocular features were not provided in detail [23].

The two frameshift variants in exon 1, c.23_32del p.(Gln8Leufs*86) and c.191del p.(Ala64Valfs*33), introduce a premature termination codon (PTC) at least 280 nucleotides after the start codon. These PTCs are predicted to be sensitive to nonsense-mediated decay (NMD) [25,26], leading to degradation of mRNA and an absence of the MYF5 protein.

The missense variant p.(Arg95Cys) forms a full-length protein and has been associated with EORVA in two members of one Yemeni family [20,24]. In vitro and in silico assays reported that mutant MYF5 is mislocalised to the cytoplasm and has lost its DNA binding ability [20]. Even if the consequence of the mutation on the protein differs between null or truncated protein with an alternative C-terminus, only a few EORVA patients have been described and a larger cohort of patients are required for more significant genotype–phenotype correlation.

Several mouse models have shown that rib morphogenesis is indirectly affected by *Mrf4* and *Myf5* expression via fibroblast growth factor (Fgf) and platelet-derived growth factor (Pdgf) mediation in myotome–sclerotome interactions. *Myf5/Mrf4* activation in hypaxial myotome signals to the adjacent sclerotome using Pdgf and Fgf to promote rib and vertebrae formation [14,27,28]. Mice lacking *Pdgfra*, the gene encoding the Pdgf receptor, have severe rib anomalies [29]. In addition, insertion of *Pdgfra* cDNA into the *Myf5* locus results in increased rib and vertebral development [14]. The genes encoding MRF4 and MYF5 are closely linked, with approximately 8.5kb separating their coding sequences on human chromosome 12 [30,31,32], and *cis*-acting interaction of *MRF4* negatively impacts expression of adjacent *MYF5* [17,18]. A *Mrf4* knockout mouse model exhibits rib defects and lacks *Myf5* expression. Rib anomalies are more severe in compound heterozygote *Myf5/Mrf4* than *Mrf4* knockout models, and the most severe are in *Myf5* homozygous mutants [15,17,33] (Table 4). These mouse models did not examine the influence of *Myf5/Mrf4* on extraocular muscle function. Rib development is highly sensitive to quantitative difference in MYF5 function [15,17].

After a genetic diagnosis, detailed phenotyping is important to assess for features that can aid diagnosis and provide clinical evidence in support of variant pathogenicity. Our cases presented with a presumptive diagnosis of CFEOM due to characteristic features of ophthalmoplegia and ptosis, however, radiology revealed scoliosis in all three patients, which revised the diagnosis to EORVA. Recognition of these additional features allows for a multi-disciplinary approach to provide the best possible care for the patients in the long-term.

## 5. Conclusions

In conclusion, we report three siblings of consanguineous parents with a novel homozygous variant c.596dupA p.(Asn199Lysfs*49) in exon 3 of *MYF5* associated with EORVA, a newly recognised syndrome easily mistaken for CFEOM, which has a different pathogenesis and systemic implications. With four EORVA-associated variants now discovered, it is important to perform genetic testing on patients with external ophthalmoplegia with and without extra-ocular features. Our family had no extra-ocular abnormalities on clinical examination, but genetic results prompted further radiological investigation, revealing scoliosis in all affected members. Patients with signs of ocular fibrosis, especially with a family history of ophthalmoplegia should undergo genetic testing and be referred to paediatric services for a full work up.

## Figures and Tables

**Figure 1 genes-15-00699-f001:**
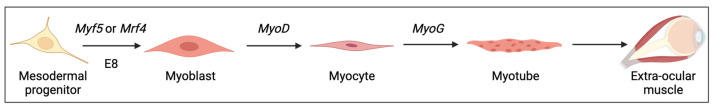
The role of myogenic transcription factors in extraocular myogenesis in mice. Extraocular muscles (EOM) are derived from cranial mesoderm progenitors. Expression of either *Myf5* or *Mrf4* is required for EOM progenitor cells to acquire their myogenic fate. *Myf5* or *Mrf4* activates *MyoD*, which in turn activates *MyoG* and EOM differentiation. Created with Biorender.

**Figure 2 genes-15-00699-f002:**
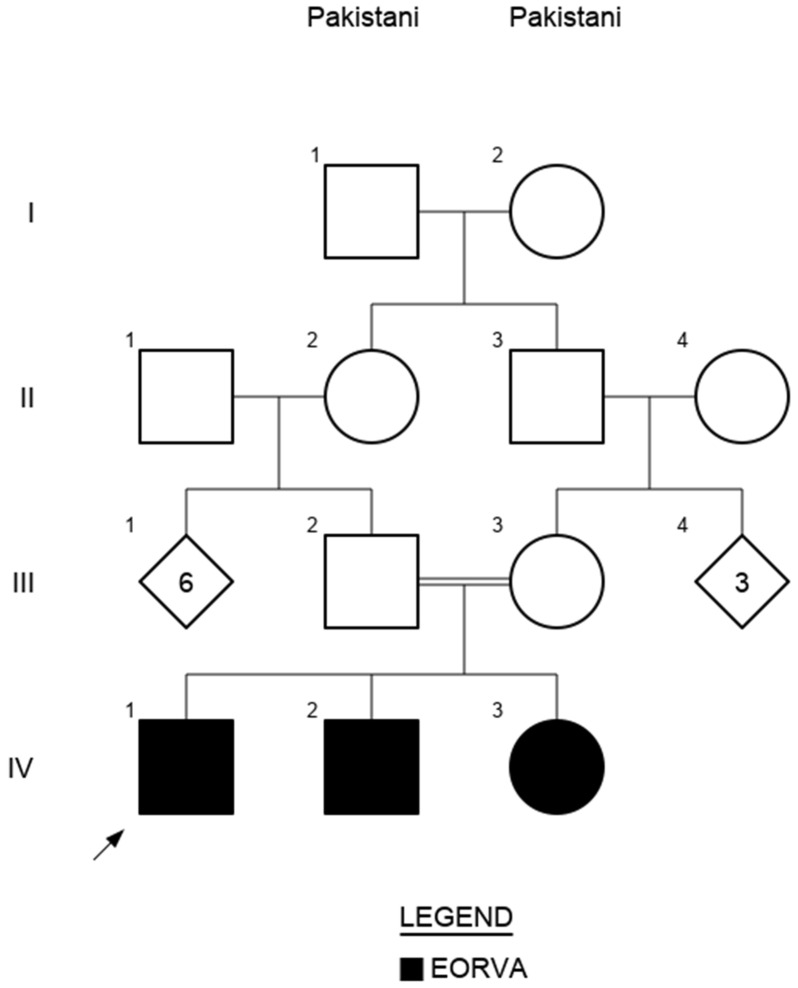
Family pedigree—three siblings from a consanguineous Pakistani family are affected with external ophthalmoplegia, with vertebral and rib anomalies. Square symbols indicate males; circles indicate females. Diamonds represent either gender. The number inside the shape is the number of individuals. Filled symbols are affected individuals. The black arrow indicates the proband.

**Figure 3 genes-15-00699-f003:**
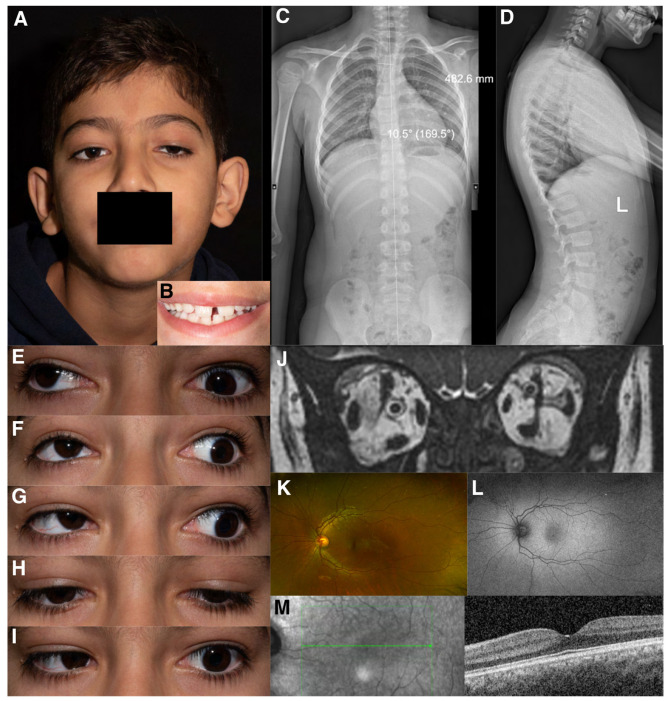
(**A**) Facial photo; (**B**) dental photo; (**C**) Patient IV-1 skeletal X-ray, antero-posterior view- mild thoracic scoliosis centred at T6-T7 concave to the left; (**D**) Patient IV-1 skeletal X-ray, lateral (L) view; gaze position photos (N.B. not all positions of gaze were obtained due to ptosis obscuring eye position); (**E**) dextro-elevation; (**F**) direct elevation; (**G**) laevo elevation; (**H**) primary position; (**I**) laevo version; (**J**) MRI orbits Patient IV-1: left medial rectus, left superior rectus, and left superior oblique muscles smaller comparing to respective EOMs in the right eye; (**K**) colour fundus photo; (**L**) autofluorescence photo; (**M**) macular OCT.

**Figure 4 genes-15-00699-f004:**
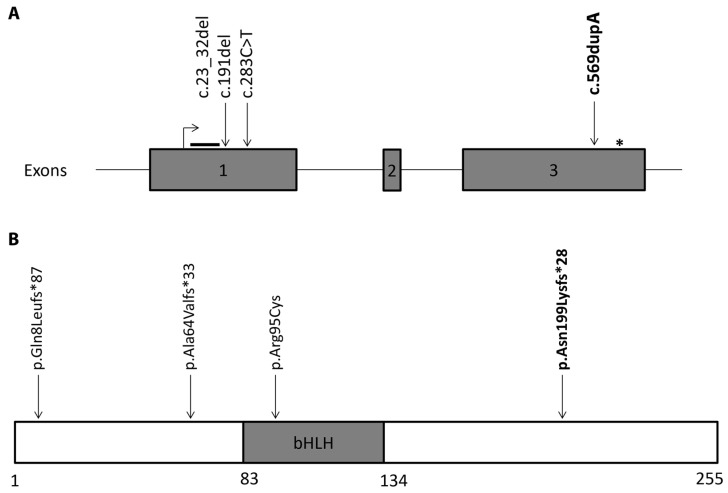
Mutational spectrum of *MYF5* related external ophthalmoplegia, with vertebral and rib anomalies. (**A**) Variants previously described depicted across exon 1 of *MYF5* gene (NM_005593.3), while the novel disease-causing variant reported in this study is located in exon 3. (**B**) Amino acid changes mapped across MYF5 transcription factor (NP_005584, UniProt P13349); bHLH-basic helix–loop–helix protein domain (amino acid residue 83 to 134) binding DNA. The novel disease-causing variant reported in this study is indicated in bold. Asterisk (*) indicates premature termination (stop) codon.

**Table 1 genes-15-00699-t001:** Similarities and differences between CFEOM and EORVA. CFEOM: congenital fibrosis of extraocular muscles; EORVA: external ophthalmoplegia with rib and vertebral anomalies. Phenotypic differences between CFEOM and EORVA are highlighted in bold.

Condition	CFEOM	EORVA
Pathophysiology	Cranial innervation disorder	Muscle disorder
Ocular features	Non-progressive ophthalmoplegia +/− ptosis	Non-progressive ophthalmoplegia +/− ptosis
Extra-ocular features	**Neurodevelopmental anomalies** **Brain anomalies** **Limb anomalies**	**Rib defects** **Vertebral defects** **Scoliosis** **Torticollis**
Inheritance	Autosomal recessive or autosomal dominant	Autosomal recessive
Causative genes	*COL25A1*; *KIF21A2*; *PHOX2A*; *TUBA1A*; *TUBB2B*; *TUBB3*	*MYF5*

**Table 2 genes-15-00699-t002:** Orthoptic and ophthalmic examination of Patient IV-1, Patient IV-2, and Patient IV-3. Abbreviations: M: months; Y: years; BC-RVA: best corrected right visual acuity (logMAR); BC-LVA: best corrected left visual acuity (logMAR); R: right eye; L: left eye; LXT: left exotropia; R/AXT: right/alternating exotropia; r/o: restriction of; B/L: bilateral.

ID	Age at Presentation	Age at Last Exam	BC-RVA	BC-LVA	Refractive Error	Strabismus	Torticollis	Ocular Motility	Ptosis	Anterior Segment	Posterior Segment
Pt IV-1	6M	8Y	0.00	0.30	R: +1.00/−0.50 × 180 L: +1.75/−1.75 × 180	Constant LXT	Chin elevation	B/L marked r/o elevation B/L small r/o depression R minimal r/o abduction R moderate r/o adduction L minimal r/o adduction L small r/o abduction	Mild B/L	Normal	Normal
Pt IV-2	23M	7Y	0.00	0.08	R: +1.50DS L: +0.75DS	Constant R/AXT	Nil	B/L moderate r/o elevation B/L minimal r/o adduction	Mild B/L	Normal	Normal
Pt IV-3	4M	4Y	0.24	0.24	R: −1.50DS L: −0.50/−1.00 × 170	Intermittent Distance LXT	Nil	B/L mild r/o elevation B/L slight r/o depression B/L full abduction and adduction	MildR	Normal	Normal
	Pt IV-1				Pt IV-2			Pt IV-3			
	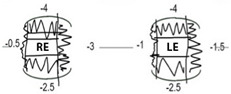				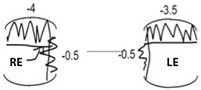			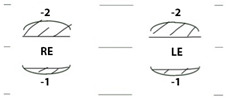			

**Table 3 genes-15-00699-t003:** Genotype–phenotype correlation in *MYF5* variants. Abbreviations: EOM: extraocular muscles; R: right eye; L: left eye; XT: exotropia; HoT: hypotropia; HT: hypertropia; +: mild; ++: moderate; +++: severe; ND: no data; ✔: present; ✘ absent.

Reference	Homozygous *MYF5* Variant	MYF5 Protein	Patient ID	External Ophthalmoplegia	Ptosis	Strabismus	Scoliosis	Torticollis	Ribs Anomalies	Vertebral Anomalies
[20]	c.23_32del	p.Gln8Leufs*86	I-1	✔	✔ R>L	XT++, HoT R+	Cervical scoliosis	✔	✔	✔
			I-2	✔	✔ R>L	XT++	ND	✔	✔	ND
			I-3	✔—EOMs hypoplastic to absent	✔ R>L	HoT++	Cervical and thoracic scoliosis	✔	✔	✔
[23]	c.191delC	p.Ala64Valfs*33	II-1	ND	✔	ND	✔	ND	✔	✘
[20,24]	c.283C>T	p.Arg95Cys	III-1	✔	✔ L	XT+++ HoT L+	ND	possibly	ND	ND
			III-2	✔	✘	XT+	Lumbar scoliosis	ND	ND	ND
This study	c.596dup	p.Asn199Lysfs*49	IV-1	✔—EOMs L hypoplastic comparing to R	✔	XT L+	Thoracic scoliosis	✘	✘	✘
			IV-2	✔	✔	XT R+,HT R +	Thoracic and lumbar scoliosis	✘	✘	✘
			IV-3	✔	✔ R	XT L+ HT L+	Lumbar scoliosis	✘	✘	✘

**Table 4 genes-15-00699-t004:** Genotype–phenotype correlation in mice models with rib morphogenesis defects; ND—no data; wt—wild type.

Genotype Mice Model	Rib Cage Defects	Vertebrae Defects
*Mrf4^−/−^* [34]	Ribs not attached to the sternum Truncation of ribs Bifurcation and fusion of adjacent ribs Irregular sternum ossification	ND
*Mrf4^−/+^Myf5^−/+^* [17]	Ribs not attached to the sternum Truncation of ribs: shorter vs *wt/Mrf4^-/-^*; longer vs. *Myf5^-/-^* Bifurcation and fusion of adjacent ribs Irregular sternum ossification	ND
*Myf5^−/−^* [15]	Absence of the distant parts of the ribs Complete ossification of the sternum Lethal immediately postnatally (inability to breath)	ND
*Pdgfra^−/−^* [29]	Ribs mostly attached to the sternum Bifurcation and fusion of adjacent ribs Irregular and shorter sternum	Structural anomalies of cervical and thoracic vertebrae Spina bifida Anomalies of spinal column curvature

## Data Availability

Data are contained within the article.

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
