# Peer review of "Identification of a Novel Frameshift Variant in MYF5 Leading to External Ophthalmoplegia with Rib and Vertebral Anomalies"

_genes, 2024, doi:10.3390/genes15060699_

Round 1

Reviewer 1 Report

Comments and Suggestions for Authors

Ocieczek et al. reported a new form of mutation in the MYF5 gene that leads to external ophthalmoplegia with rib and vertebral anomalies (EORVA). Three siblings from the same family are reported having external ophthalmoplegia, ptosis and scoliosis symptoms while their parents were unaffected. Through whole genome sequencing, they identified the frameshift mutation in the MYF5 gene while all three siblings were found homozygous for this variant. 

1. Fig. 1, I would suggest using a developing eye diagram instead of an embryo one, kinda of hard to tell what you're pointing at. Also please include the embryonic stage of extra-ocular muscle development. 

2. Fig. 2, what does the "6" and "3" mean for the unknown individuals in generation III? 

3. Fig. 3, for J-M could you put scale bars for those?

4. Fig. 4, maybe use an arrow or arrowhead to indicate the relative localization of the newly discovered variant in exons of the MYF5 gene.

5. Table 2, maybe use symbols instead of YES/NO? 

6. Based on the content from 197-210, you could make another table of animal models showing related phenotypes. Include both gene-related and symptoms-related. 

7. See whether there are some myoblast/myocyte studies (cell culture experiments) that investigate the roles of MRF4/MYF5 in cell differentiation, they should be a good range of references. 

Comments on the Quality of English Language

The overall quality of this manuscript is good. 

Author Response

Reviewer 1:

Ocieczek et al. reported a new form of mutation in the MYF5 gene that leads to external ophthalmoplegia with rib and vertebral anomalies (EORVA). Three siblings from the same family are reported having external ophthalmoplegia, ptosis and scoliosis symptoms while their parents were unaffected. Through whole genome sequencing, they identified the frameshift mutation in the MYF5 gene while all three siblings were found homozygous for this variant. 

  1. Fig. 1, I would suggest using a developing eye diagram instead of an embryo one, kind of hard to tell what you're pointing at. Also please include the embryonic stage of extra-ocular muscle development. 

We have edited the diagram (Figure 1). However, as the embryonic stages of extra-ocular muscle development vary depending on the source, we have not labelled all of the embryonic stages on the figure. We have removed the image on an embryo and labelled E8.

2.Fig. 2, what does the "6" and "3" mean for the unknown individuals in generation III? 

The numbers inside the shapes are condensed family members, with the number being the number of family members/siblings represented by the shape. The Figure 2 legend has been updated as follow:

Figure 2. Family pedigree - three siblings from a consanguineous Pakistani family are affected with external ophthalmoplegia, vertebral and rib anomalies. Square symbols indicate males; circles indicate females. Diamonds represent either gender. The number inside the shape is the number of individuals. Filled symbols are affected individuals.

  1. Fig. 3, for J-M could you put scale bars for those?

Thank you for the comment. Scale bars are not commonly included for MRI or ocular images.

  1. Fig. 4, maybe use an arrow or arrowhead to indicate the relative localization of the newly discovered variant in exons of the MYF5 gene.

Please see the novel variant is indicated in bold in Figure 4A, and this is outlined in the legend.

  1. Table 2, maybe use symbols instead of YES/NO? 

We have changed Yes/No for tick and cross symbols.

  1. Based on the content from 197-210, you could make another table of animal models showing related phenotypes. Include both gene-related and symptoms-related. 

The following table is added to the manuscript from line 237-line 240

Table 4. Genotype-phenotype correlation in mice models with rib morphogenesis defects; ND - no data; wt - wild type.

Genotype mice model

Rib cage defects

Vertebrae defects

Mrf4-/- [34]

Ribs not attached to the sternum

Truncation of ribs

Bifurcation and fusion of adjacent ribs

Irregular sternum ossification

ND

Mrf4-/+Myf5-/+ [17]

Ribs not attached to the sternum

Truncation of ribs: shorter vs wt/Mrf4-/- ; longer vs. Myf5-/-

Bifurcation and fusion of adjacent ribs

Irregular sternum ossification

ND

Myf5-/- [15]

Absence of the distant parts of the ribs

Complete ossification of the sternum

Lethal immediately postnatally (inability to breath)

ND

Pdgfra-/- [29]

Ribs mostly attached to the sternum

Bifurcation and fusion of adjacent ribs

Irregular and shorter sternum

Structural anomalies of cervical and thoracic vertebrae

Spina bifida

Anomalies of spinal column curvature

  1. See whether there are some myoblast/myocyte studies (cell culture experiments) that investigate the roles of MRF4/MYF5 in cell differentiation, they should be a good range of references.

This manuscript is based on a clinical case report, and we decided not to go into further discussions on myoblast studies/experiments to keep the manuscript concise. We have focused on published clinical cases and animal models.

Reviewer 2 Report

Comments and Suggestions for Authors

Introduction

Line  33, please add a reference about structure and functio od EOM 

case report

line 85 

parents are reported to be normal;  would it be possible to report some more details (x-rays, eye examination), confirming complete absense of any skeletal and ocular problems?

authors report in detail previously published cases;  could they add a comment to specify if in carriers any clinicsl sign was really searched and eventually found?

Author Response

Reviewer 2:

Line 33, please add a reference about structure and function of EOM 

We added references 4 and 5 to the manuscript.

Line 85 

Parents are reported to be normal; would it be possible to report some more details (x-rays, eye examination), confirming complete absence of any skeletal and ocular problems?

The parents were not formally tested for any ocular problems and in the absence of ocular or skeletal symptoms they did not have any imaging.

Authors report in detail previously published cases; could they add a comment to specify if in carriers any clinical sign was really searched and eventually found?

Carrier phenotype was not detailed in the papers cited. They only mentioned that parents were ‘unaffected’.

Reviewer 3 Report

Comments and Suggestions for Authors

Multi Digital Publishing Institute (MDPI): Genes-3010922

Title: Case Repor

Identification of a novel frameshift variant in MYF5 associated with external ophthalmoplegia, rib and vertebral anomalies

Review request received on 05/09/2024.

Overall comments:

This is a clinical case report on three siblings born to consanguineous parents of Pakistani descent who were originally referred to the genetic eye disease clinic with a presumptive clinical diagnosis of congenital fibrosis of extraocular muscles (CFEOM), and they were identified to have a homozygous novel variant in the MTF5 gene via genetic analyses, which is the known causal gene for external ophthalmoplegia, rib, and vertebral anomalies (EORVA).  Then, additional evaluation revealed vertebral and other bony anomalies, consistent with the diagnosis of EORVA. 

It should be noted that consanguinity is a well-known risk factor for uncovering an autosomal recessive condition. 

Overall, a case of EORVA is very rare and interesting.  The manuscript is fairly well-written; however, some re-arrangement and revision (editing) of the content would improve the manuscript much more aligned to be a monogenic clinical case report and would improve its readability. 

In addition, there should be a good teaching point included in the presentation for such a clinical case, not just presenting a novel genetic discovery alone. 

Very minor comment about the title: when one uses “associated” vs “causal”, the relationship between the variant and the disease becomes somewhat vague as in genome wide association studies.  Thus, recommend replacing “associated” with something more concrete in describing the relationship such as “caused or causal”.

           First of all, in a clinical genetic report, details about the condition/disorder should be presented first instead of going into details about the biology of muscle development, and of genes and their roles etc. so that the readers became interested as a clinical case report, focusing as a patient-oriented presentation. 

           Furthermore, it is also important to have a section on management of EORVA, and possibly about CFEOM as well.  In clinical practice, once a diagnosis is confirmed, management of patients becomes very important in preventing and minimizing the morbidity or mortality associated with the disease.  It is often the case that there may be no curable therapy in many genetic disorders, but it is still important to improve the lives of patients with some type of therapeutic management.

           Although it is always important and desirable to have functional studies for novel variants, for this particular variant, due to its rarity and a newly created premature stop codon, the variant can be assumed to have a detrimental effect on the protein.  Thus, the variant can be considered as the “likely causal gene variant” for these patients.

Finding additional patients worldwide with the same genotype and phenotype or even within the unrelated Pakistani community would definitely strengthen this claim though future functional studies would be great and desirable despite having some data from mice studies. 

Please see the below for additional comments. 

Specific comments:

1. Introduction

           This manuscript is a clinical case report of a monogenic disorder; therefore, it would be more desirable to start with the description of the disorder (EORVA) first as mentioned above, before getting into the developmental biology of the genes involved in the introduction, especially much of it pertains to mice development and studies, not in humans.

Therefore, the condition, external ophthalmoplegia with rib and vertebral abnormalities (EORVA) should be presented in the first paragraph.  Known representative phenotypes as well as epidemiological information should be included. 

The clinical information would be especially informative and important for physicians or ophthalmologists who actually may see patients with this condition in the future.

Then, explain how EORVA is different from congenital fibrosis of extraocular muscles (CFEOM) as explained in Lines 61-71.  If they are very similar as the authors mentioned, it would be important to emphasize some specific similarities and differences between two disorders so that clinicians would pay attention, presenting them as a teaching point of this case report. 

It may also be helpful to have a table comparing the two conditions so that other physicians who have not seen either condition will be able to understand better from the aspects of ophthalmology as well as other physical features.

My understanding is that external ophthalmoplegia, EORVA presents as weakness of the eye muscles due to aberrant development with deficient MYF5, but congenital fibrosis of extraocular muscles (CFEOM) is due to a neurological abnormality, more of a nervous system disorder affecting the use of extraocular muscles.  Thus, although they may appear similar, each pathophysiology seems very different so these points should be stressed.     

Once these clinical disorders are discussed, pathophysiology and developmental biology should be discussed next.  Understanding the genes and their pathophysiology and underlying biological mechanisms are important in monogenic cases, but this should be presented after disorder(s) and differential diagnoses have been discussed first. 

2. Case report:

The authors presented a brief description of each child, and some background information is provided.   

Line 118: It was a bit confusing how the consideration of MYF5 came about.  Later it became clearer that the genetic testing was performed first, and because of the finding of the novel homozygous variant in MYF5, then the diagnosis of EORVA was considered. 

Typically, clinical geneticists who examine patients may notice other features to arrive at a potential clinical diagnosis first or differential diagnoses, but does this mean that ophthalmologists agreed about the diagnosis of CFEOM?  Or since multiple family members had similar condition that hereditary condition was considered? 

Recommend writing about the authors’ or the ophthalmologists’ thought processes during evaluation.  Furthermore, detailed notes on physical examination if available should be presented even if they are negative in addition to ophthalmological findings, not just radiographical findings. 

When the eye issues were noted, was there anything done prior to the visit to the clinic?

3. Genetic Testing

The authors should always make a comment on the known limitations of the genetic testing methods utilized.  The exact targets such as exons, exon-intron boundaries (number of nucleotide information if possible) and 5’-UTRs etc., as well as the detection capabilities of any copy number variants (CNVs).  If a next-generation sequencing (NGS) method was used, then, how much was the depth?

What were the fourteen genes associated with eye movements?  It would be helpful for the readers to have a list (maybe in an appendix table) so that it is clear to the readers what exactly was performed.  There are so many biochemical genetic disorders which present with eye movement abnormalities as well though other types of test are needed for these in addition. 

Each country has its own regulatory framework on the clinical use of genetic tests, but in the United States, commercial laboratories which are Clinical Laboratory Improvement Amendments (CLIA) compliant would need to be testing research results for confirmation in order for the genetic finding to be used in the clinical care of patients.  Although for this case report, it was not clear if the results were shared with the patients at the time of this report is written though if the patients agreed to be written about them, they probably were told about the results?

Line 144:  It may be better to add the nucleotide duplicated although it may be clear to the laboratory-related personnel, but the readers may not be familiar with the MYF5 gene and its sequence.   MYF5, c.596dupA, p.(Asn199Lysfs*49).  Potentially, insertion of a nucleotide can be used in this case unless a mechanism has been known for this occurrence (repeated) of duplication in this gene.

Please briefly discuss management strategies for patients with EORVA, not only for the eyes, but also for skeletal abnormalities.  Some information about management of eye movement abnormalities may also be helpful. 

Recommend adding a brief case summary section prior to the discussion section so that the readers are clear about the findings from the case or the definitive diagnosis obtained.  (4. Summary results?)

4. Discussion

Here also, since this is a case report, it should start with the patients’ information rather than the description of the disease (patient-oriented presentation would be recommended).

e.g.,

Here, we present the three siblings who were originally referred with a presumptive diagnosis of …., was identified with EORVA through the identification of a novel homozygous genetic variant….  After the genetic results, the patients were evaluated for …., and found to have ….

Then, recommend stating the description of the variant next. 

Is any functional information on the domains or exons of the protein MYF5 available?  If this is a transcription factor.. binding etc.? (may be add to the diagram?).

Any deletion, insertion, or duplication will likely result in a change in nucleotide sequence as well as protein sequence (e.g., frameshift as in the title).  In addition, when a premature stop codon is generated, mRNAs with premature stop codons are often cleared from the cell via nonsense-mediated mRNA decay as the authors mentioned, with some exceptions. 

For this reason, any variant with a premature stop codon can often be considered as “likely pathogenic”.

Lines 191-196: This section should also be included in the genetic testing section.

As the authors state, per American College of Medical Genetics (ACMG), this variant is considered “likely pathogenic” even without a detailed functional result although functional studies would be helpful, and definitely, can confirm the pathogenicity of the variant or finding other families with the same genotype-phenotype. 

Population Data: PM2 (extremely low frequency in a population database such as gnomAD).

Since this novel variant results in generating a premature stop codon, the below may potentially be applicable.

PVS1: Predicted null variant in a gene where LOF is a known mechanism of disease.  Agree with the authors.

At the end, it was clear that no one seemed to have noticed the rib anomalies in the patients, but this should also be stated in the case report above that someone did an attempt to examine these patients’ physical features.  If it is noticeable by inspection, this should be noted there. 

5. Conclusion

Here please also write in the patient-oriented way. 

Example:

In conclusion, we report three siblings of consanguineous parents identified with a novel homozygous variant in MYF5, c.596dupA, p.(Asn199Lysfs*49) in exon 3, …

Figure 3:

Has there been a case with dental anomaly noted with EORVA?  Please comment why this picture was included.  It is often seen in a number of genetic conditions with dental anomalies with skeletal anomalies.

Table 2:

Abbreviations may be more suitable below the table.

It was hard for me to refer to the notations while looking through the table (minor point).

Please correct some irregularities, “no” to “No”.

Please consider adding following if feasible:

1.     Comparison table of EORVA and CFEOM.

2.     14 genes related to eye movement which were targeted (could be added as an appendix table).

Thank you very much for allowing me to review this manuscript. 

Sincerely,

Author Response

Reviewer 3:

Overall comments:

This is a clinical case report on three siblings born to consanguineous parents of Pakistani descent who were originally referred to the genetic eye disease clinic with a presumptive clinical diagnosis of congenital fibrosis of extraocular muscles (CFEOM), and they were identified to have a homozygous novel variant in the MTF5 gene via genetic analyses, which is the known causal gene for external ophthalmoplegia, rib, and vertebral anomalies (EORVA). Then, additional evaluation revealed vertebral and other bony anomalies, consistent with the diagnosis of EORVA. 

It should be noted that consanguinity is a well-known risk factor for uncovering an autosomal recessive condition. 

Overall, a case of EORVA is very rare and interesting.  The manuscript is fairly well-written; however, some re-arrangement and revision (editing) of the content would improve the manuscript much more aligned to be a monogenic clinical case report and would improve its readability. 

In addition, there should be a good teaching point included in the presentation for such a clinical case, not just presenting a novel genetic discovery alone. 

We believe that the teaching point in the case report is pointed out in the conclusion (line 251-257):

“With four EORVA-associated variants now discovered, it is important to perform genetic testing on patients with external ophthalmoplegia with and without extra-ocular features. Our family had no extra-ocular abnormalities on clinical examination, but genetic results prompted further radiological investigation revealing scoliosis in all affected members. Patients with signs of ocular fibrosis, especially with a family history of ophthalmoplegia should undergo genetic testing and be referred to paediatric services for a full work up.”

Very minor comment about the title: when one uses “associated” vs “causal”, the relationship between the variant and the disease becomes somewhat vague as in genome wide association studies.  Thus, recommend replacing “associated” with something more concrete in describing the relationship such as “caused or causal”.

We have changed ‘associated with’ to ‘leading to’ in the title as follows: (line 2)

“Identification of a novel frameshift variant in MYF5 leading to external ophthalmoplegia, rib and vertebral anomalies.”

First of all, in a clinical genetic report, details about the condition/disorder should be presented first instead of going into details about the biology of muscle development, and of genes and their roles etc. so that the readers became interested as a clinical case report, focusing as a patient-oriented presentation. 

Thank you for your suggestion. We believe that the current outline of the introduction, explaining the mechanism and pathophysiology behind the condition is important for the reader to understand the correlation between the ocular and systemic features of the syndrome. This is not well known or described condition; therefore, we believe this outline helps clinicians in familiarising themselves with this new entity by understanding the pathology and genetic nature of the condition first.

           Furthermore, it is also important to have a section on management of EORVA, and possibly about CFEOM as well.  In clinical practice, once a diagnosis is confirmed, management of patients becomes very important in preventing and minimizing the morbidity or mortality associated with the disease.  It is often the case that there may be no curable therapy in many genetic disorders, but it is still important to improve the lives of patients with some type of therapeutic management.

Management paragraph was added to the introduction as suggested: (line 76-83)

“The characteristic ocular features are similar in both EORVA and CFEOM. Overlooking mild systemic features which differ in both these conditions can lead to misdiagnosis and suboptimal management (Table 1). Ocular management in EORVA and CFEOM include monitoring and correction of refractive error and amblyopia; in some cases, surgery to correct ptosis or extraocular muscles alignment should be considered. Management of systemic complications will vary depending on presenting features and include multi-disciplinary management and the involvement of a paediatrician.”

           Although it is always important and desirable to have functional studies for novel variants, for this particular variant, due to its rarity and a newly created premature stop codon, the variant can be assumed to have a detrimental effect on the protein.  Thus, the variant can be considered as the “likely causal gene variant” for these patients.

Finding additional patients worldwide with the same genotype and phenotype or even within the unrelated Pakistani community would definitely strengthen this claim though future functional studies would be great and desirable despite having some data from mice studies. 

Agreed.

Please see the below for additional comments. 

Specific comments:

  1. Introduction

           This manuscript is a clinical case report of a monogenic disorder; therefore, it would be more desirable to start with the description of the disorder (EORVA) first as mentioned above, before getting into the developmental biology of the genes involved in the introduction, especially much of it pertains to mice development and studies, not in humans.

Therefore, the condition, external ophthalmoplegia with rib and vertebral abnormalities (EORVA) should be presented in the first paragraph.  Known representative phenotypes as well as epidemiological information should be included. 

We prefer to keep the current outline as explained above. There are no epidemiological studies, as this syndrome has been previously described only in six patients in four unrelated families.

The clinical information would be especially informative and important for physicians or ophthalmologists who actually may see patients with this condition in the future.

Then, explain how EORVA is different from congenital fibrosis of extraocular muscles (CFEOM) as explained in Lines 61-71.  If they are very similar as the authors mentioned, it would be important to emphasize some specific similarities and differences between two disorders so that clinicians would pay attention, presenting them as a teaching point of this case report. 

It may also be helpful to have a table comparing the two conditions so that other physicians who have not seen either condition will be able to understand better from the aspects of ophthalmology as well as other physical features.

Thank you for the suggestion: We have added Table 1(line 84-88) comparing CFEOM and EORVA and also pointed out in the text that ocular features are very similar and systemic features distinguish those two entities:

Table 1. Similarities and differences between CFEOM and EORVA. CFEOM: congenital fibrosis of extraocular muscles; EORVA external ophthalmoplegia with rib and vertebral anomalies. Phenotypic differences between CFEOM and EORVA are highlighted in bold.

Condition

CFEOM

EORVA

Pathophysiology

Cranial innervation disorder

Muscle disorder

Ocular features

Non-progressive ophthalmoplegia

+/- ptosis

Non-progressive ophthalmoplegia

+/- ptosis

Extra-ocular features

Neurodevelopmental anomalies

Brain anomalies

Limb anomalies

Rib defects

Vertebral defects

Scoliosis

Torticollis

Inheritance

Autosomal recessive or autosomal dominant

Autosomal recessive

Causative genes

COL25A1; KIF21A2; PHOX2A; TUBA1A; TUBB2B; TUBB3

MYF5

My understanding is that external ophthalmoplegia, EORVA presents as weakness of the eye muscles due to aberrant development with deficient MYF5, but congenital fibrosis of extraocular muscles (CFEOM) is due to a neurological abnormality, more of a nervous system disorder affecting the use of extraocular muscles.  Thus, although they may appear similar, each pathophysiology seems very different so these points should be stressed.   

Different pathophysiology stressed in the introduction and now also in Table 1.

Once these clinical disorders are discussed, pathophysiology and developmental biology should be discussed next.  Understanding the genes and their pathophysiology and underlying biological mechanisms are important in monogenic cases, but this should be presented after disorder(s) and differential diagnoses have been discussed first.

  1. Case report:

The authors presented a brief description of each child, and some background information is provided.   

Line 118: It was a bit confusing how the consideration of MYF5 came about. Later it became clearer that the genetic testing was performed first, and because of the finding of the novel homozygous variant in MYF5, then the diagnosis of EORVA was considered.

Typically, clinical geneticists who examine patients may notice other features to arrive at a potential clinical diagnosis first or differential diagnoses, but does this mean that ophthalmologists agreed about the diagnosis of CFEOM? Or since multiple family members had similar condition that hereditary condition was considered? 

Recommend writing about the authors’ or the ophthalmologists’ thought processes during evaluation.  Furthermore, detailed notes on physical examination if available should be presented even if they are negative in addition to ophthalmological findings, not just radiographical findings. 

When the eye issues were noted, was there anything done prior to the visit to the clinic?

In the genetic clinic, the likely diagnosis of congenital fibrosis of extraocular muscles (CFEOM) was made first, with suspected autosomal recessive inheritance as the proband had an affected younger brother and unaffected consanguineous parents.

Once the homozygous MYF5 variant was found, the systemic features were further investigated as described in the case presentation section.

  1. Genetic Testing

The authors should always make a comment on the known limitations of the genetic testing methods utilized.  The exact targets such as exons, exon-intron boundaries (number of nucleotide information if possible) and 5’-UTRs etc., as well as the detection capabilities of any copy number variants (CNVs).  If a next-generation sequencing (NGS) method was used, then, how much was the depth?

 What were the fourteen genes associated with eye movements?  It would be helpful for the readers to have a list (maybe in an appendix table) so that it is clear to the readers what exactly was performed.  There are so many biochemical genetic disorders which present with eye movement abnormalities as well though other types of test are needed for these in addition. 

Each country has its own regulatory framework on the clinical use of genetic tests, but in the United States, commercial laboratories which are Clinical Laboratory Improvement Amendments (CLIA) compliant would need to be testing research results for confirmation in order for the genetic finding to be used in the clinical care of patients.  Although for this case report, it was not clear if the results were shared with the patients at the time of this report is written though if the patients agreed to be written about them, they probably were told about the results?

From line 146-169 genetic test was more detailed as suggested:

Informed consent was obtained from all subjects involved in the study through the Genetic Study of Inherited Eye Disease (REC reference 12/LO/0141). A clinical exome (Agilent SureSelect Focused Exome +1 capture) for patient IV-1 was performed on the Illumina NextSeq 500 platform, with sequence data generated across the full capture region of greater than 5,000 genes. Next generation sequencing analysis was then performed for a virtual panel of coding exons (-/+ 20 bp) of 14 genes associated with eye movement disorders (EMD_v2 panel, North East Thames Regional Genetics Laboratory: CHN1, COL25A1, DCC, FRMD7, HOXA1, HOXB1, KIF21A, MAFB, PHOX2A, ROBO3, SALL1, SALL4, TUBB2B, TUBB3 ). Larger insertion/deletion mutations and copy number variants were analysed using ExomeDepth. Variants were filtered according to minor allele frequency (>2%) from 1000G, ExAC or EVS databases. No pathogenic or likely pathogenic variants were identified. Variants in non-coding regions that could affect gene expression could not be excluded

Both brothers IV-1 and IV-2 were subsequently recruited to the Genomics England 100,000 Genomes Project together with their unaffected parents for whole genome sequencing as previously described [19], [20]. Both brothers were found to be homozygous, and the unaffected parents heterozygous carriers for a novel duplication c.596dupA in the MYF5 gene resulting in a frameshift variant p.(Asn199Lysfs*49). Targeted sequence analysis of the MYF5 variant confirmed the 100,000 Genome Project findings. The third affected sibling, IV-3, underwent familial MYF5 testing and was found to be homozygous for the same variant.  In the gnomAD population database this variant was found at heterozygous state in two individuals (f = 0.00000124) and has not been described before at homozygous state.

Line 144:  It may be better to add the nucleotide duplicated although it may be clear to the laboratory-related personnel, but the readers may not be familiar with the MYF5 gene and its sequence.   MYF5, c.596dupA, p.(Asn199Lysfs*49).  Potentially, insertion of a nucleotide can be used in this case unless a mechanism has been known for this occurrence (repeated) of duplication in this gene.

c.596dupA was added to the manuscript as suggested.

Please briefly discuss management strategies for patients with EORVA, not only for the eyes, but also for skeletal abnormalities.  Some information about management of eye movement abnormalities may also be helpful. 

Management has been added in the introduction as described above.

Recommend adding a brief case summary section prior to the discussion section so that the readers are clear about the findings from the case, or the definitive diagnosis obtained.  (4. Summary results?)

Summary of the phenotype of probands was added in the discussion section as follows: (line 177-179)

“Here we present the three siblings with EORVA (non-progressive ophthalmoplegia, ptosis and scoliosis), without vertebrae anomalies and torticollis, caused by a novel homozygous MYF5 frameshift variant, c.596dupA, p.(Asn199Lysfs*49), in exon 3.”

  1. Discussion

Here also, since this is a case report, it should start with the patients’ information rather than the description of the disease (patient-oriented presentation would be recommended).

e.g.,

Here, we present the three siblings who were originally referred with a presumptive diagnosis of …., was identified with EORVA through the identification of a novel homozygous genetic variant….  After the genetic results, the patients were evaluated for …., and found to have ….

Summary of the phenotype of probands was added in the discussion section as follows: (line 177-179)

“Here we present the three siblings with EORVA (non-progressive ophthalmoplegia, ptosis and scoliosis), without vertebrae anomalies and torticollis, caused by a novel homozygous MYF5 frameshift variant, c.596dupA, p.(Asn199Lysfs*49), in exon 3.”

Then, recommend stating the description of the variant next. 

Is any functional information on the domains or exons of the protein MYF5 available?  If this is a transcription factor binding etc.? (may be add to the diagram?).

Figure 4 (line 185-line 193) present domain bHLH and exon 3. The legend has been updated to clarify role of bHLH domain as follows:

Figure 4. Mutational spectrum of MYF5 related external ophthalmoplegia, vertebral and rib anomalies. (A) Variants previously described depicted across exon 1 of MYF5 (NM_005593.3), while the novel disease-causing variant reported in this study is located in exon 3. (B) Amino acid changes mapped across MYF5 transcription factor (NP_005584, UniProt P13349); bHLH-basic helix-loop-helix protein domain (amino acid residue 83 to 134) binding DNA. The novel disease-causing variant reported in this study is indicated in bold.

Any deletion, insertion, or duplication will likely result in a change in nucleotide sequence as well as protein sequence (e.g., frameshift as in the title).  In addition, when a premature stop codon is generated, mRNAs with premature stop codons are often cleared from the cell via nonsense-mediated mRNA decay as the authors mentioned, with some exceptions. 

For this reason, any variant with a premature stop codon can often be considered as “likely pathogenic”.

Agreed.

Lines 191-196: This section should also be included in the genetic testing section.

As the authors state, per American College of Medical Genetics (ACMG), this variant is considered “likely pathogenic” even without a detailed functional result although functional studies would be helpful, and definitely, can confirm the pathogenicity of the variant or finding other families with the same genotype-phenotype. 

This paragraph has been moved to the Genetic testing section as suggested.

Population Data: PM2 (extremely low frequency in a population database such as gnomAD).

Since this novel variant results in generating a premature stop codon, the below may potentially be applicable.

PVS1: Predicted null variant in a gene where LOF is a known mechanism of disease.  Agree with the authors.

At the end, it was clear that no one seemed to have noticed the rib anomalies in the patients, but this should also be stated in the case report above that someone did an attempt to examine these patients’ physical features.  If it is noticeable by inspection, this should be noted there.

This is explained in the Case presentation section line 131-136:

“Therefore, following the genetic results, patients IV-1, IV-2 and IV-3 were assessed by a developmental paediatrician to look for syndromic features, in particular torticollis, scoliosis, spinal, rib cage or chest abnormalities, none of which were found on physical examination. Neurological examination and growth indices were within normal limits. Patients were referred for spinal x-rays to investigate for radiological evidence of ribcage and spinal abnormalities.”

  1. Conclusion

Here please also write in the patient-oriented way. 

Example:

In conclusion, we report three siblings of consanguineous parents identified with a novel homozygous variant in MYF5, c.596dupA, p.(Asn199Lysfs*49) in exon 3, …

Edited as suggested (line 248-251):

In conclusion, we report three siblings of a consanguineous parents with a novel homozygous variant c.596dup p.(Asn199Lysfs*49) in exon 3 of MYF5 associated with EORVA, a newly recognised syndrome easily mistaken for CFEOM, which has a different pathogenesis and systemic implications.

Figure 3:

Has there been a case with dental anomaly noted with EORVA?  Please comment why this picture was included.  It is often seen in a number of genetic conditions with dental anomalies with skeletal anomalies.

There has not been a case of dental anomaly reported with EORVA. However, only six cases have been described. As there were dental anomalies noted in one of the three siblings, we believe it is interesting to include the photography to present the full phenotype. This could be an indication to investigate dental anomalies further if more cases are reported, however, we are unable to determine whether this is part of the EORVA phenotype.

Table 2:

Abbreviations may be more suitable below the table. It was hard for me to refer to the notations while looking through the table (minor point).

The current format is the format requested by the Genes guidelines for authors.

Please correct some irregularities, “no” to “No”.

This has been corrected.

Please consider adding following if feasible:

  1. Comparison table of EORVA and CFEOM.

This has been added in the introduction as mentioned above.

  1. 14 genes related to eye movement which were targeted (could be added as an appendix table).

This has been added in the genetic testing section as presented above.
